# The structure of a hibernating ribosome in a Lyme disease pathogen

Manjuli R. Sharma [1], Swati R. Manjari [1], Ekansh K. Agrawal[1,4], Pooja Keshavan[1], Ravi K. Koripella [1,5], Soneya Majumdar[1], Ashley L. Marcinkiewicz[2], Yi-Pin Lin [2,3], Rajendra K. Agrawal [1,3] ✉ & Nilesh K. Banavali [1,3] ✉

The spirochete bacterial pathogen *Borrelia* (*Borreliella*) *burgdorferi* (*Bbu*) affects more than 10% of the world population and causes Lyme disease in about half a million people in the US annually. Therapy for Lyme disease includes antibiotics that target the *Bbu* ribosome. Here we present the structure of the *Bbu* 70S ribosome obtained by single particle cryo-electron microscopy at 2.9 Å resolution, revealing a bound hibernation promotion factor protein and two genetically non-annotated ribosomal proteins bS22 and bL38. The ribosomal protein uL30 in *Bbu* has an N-terminal α-helical extension, partly resembling the mycobacterial bL37 protein, suggesting evolution of bL37 and a shorter uL30 from a longer uL30 protein. Its analogy to proteins uL30m and mL63 in mammalian mitochondrial ribosomes also suggests a plausible evolutionary pathway for expansion of protein content in mammalian mitochondrial ribosomes. Computational binding free energy predictions for antibiotics reflect subtle distinctions in antibiotic-binding sites in the *Bbu* ribosome. Discovery of these features in the *Bbu* ribosome may enable better ribosome-targeted antibiotic design for Lyme disease treatment.

Lyme disease affects up to 14.5% of the human population worldwide[1] and is the most prevalent tick-borne disease in the Northern hemisphere, including the United States[2]. Its causative agent is the spirochete bacteria genospecies complex, *Borrelia burgdorferi* sensu lato (also known as *Borreliella burgdorferi* sensu lato or Lyme borreliae), which includes the species *Borrelia burgdorferi* sensu stricto (*Bbu*), the primary cause of human Lyme disease in North America. Elongated seasons and widened habitats for the tick vectors driven by climate change are resulting in its continued rise[3]. *Bbu* transmitted from tick saliva into human skin after the tick bite can spread and affect the human cardiovascular and nervous systems[4]. The direct economic burden of Lyme disease on the United States healthcare system alone is estimated to be a multi-billion amount each year[5].

The ribosome is an RNA-protein molecular machine that coordinates the vital process of protein synthesis in all living organisms[6].

Over two decades of detailed structural studies on various prokaryotic and eukaryotic ribosomes have clarified various aspects of the four mechanistic steps of translation – initiation, elongation, termination, and recycling[7–9]. However, even in prokaryotes, new 70S ribosome structures continue to reveal unexpected functional and compositional features, such as the discovery of structural basis of specific ribosome hibernation mechanisms[10–17] and presence of new smaller ribosomal proteins[14,18–20].

When diagnosed early in the infection, Lyme infections can be adequately treated with antibiotics, including ribosome-targeting antibiotics such as doxycycline and erythromycin[21]. Recently, hygromycin A (HygA), a ribosomal large (50S) subunit-binding antibiotic, was discovered to be extremely selective in resolving *Bbu* infections[22]. Knowledge of the structural details of the *Bbu* ribosome are therefore highly relevant for designing better biomedical interventions for Lyme disease.

[1]Division of Translational Medicine, Wadsworth Center, New York State Department of Health, Albany, NY, USA. [2]Division of Infectious Diseases, Wadsworth Center, New York State Department of Health, Albany, NY, USA. [3]Department of Biomedical Sciences, School of Public Health, University at Albany, Albany, NY, USA. [4]Present address: University of California at Berkeley, Berkeley, CA, USA. [5]Present address: Apkarian Integrated Electron Microscopy Core, Emory University, Atlanta, GA, USA. ✉e-mail: rajendra.agrawal@health.ny.gov; nilesh.banavali@health.ny.gov

In this work, we report single particle cryogenic electron microscopy (cryo-EM) structures for a hibernating *Bbu* 70S ribosome at 2.9 Å resolution and a 70S dissociation product form or assembly intermediate form of a *Bbu* 50S subunit at 3.4 Å resolution. A *Bbu* hibernation promoting factor protein (bbHPF) and two ribosomal proteins that are not annotated in the *Bbu* genome, bS22 and bL38, are found in the *Bbu* ribosome. A unique N-terminal α-helical extension in the *Bbu* ribosomal protein uL30 partially resembles the bL37 protein in mycobacterial ribosomes. The mammalian mitochondrial ribosomal proteins uL30m and mL63 are also analogous to the *Bbu* uL30 protein. These findings suggest evolution of bL37 and a shorter uL30 from a longer uL30 protein and a pathway for protein content expansion in mammalian mitochondrial ribosomes. Finally, the detailed *Bbu* 70S ribosome structure is used to obtain computational binding free energy predictions for ribosome-targeting antibiotics in clinical use for Lyme disease.

## Results and discussion

### Overall structure of the Bbu hibernating 70S ribosome

In this study, we report a cryo-EM structure for the 70S *Bbu* ribosome in its hibernating state solved at a resolution of 2.9 Å. The local resolution of the *Bbu* ribosome density is near the Nyquist limit of 2.2 Å in the core regions of the ribosome with the flexible regions such as parts of the small (30S) subunit head, the large (50S) subunit components such as uL1 stalk, the uL7/uL12-stalk base and uL9 showing lower resolution (Fig. 1a, Supplementary movie 1). The structure contains 58 resolved components (Supplementary Table 1, Fig. 1b): 3 ribosomal RNAs (23S, 5S, 16S), 1 tRNA in the E-site (Fig. 1c, d), 21 proteins in the small subunit - uS3 through bS22 and the bbHPF protein, and 33 proteins in the large subunit - uL1 through bL38 (Supplementary Fig. 1). Of these, the existence of two previously unknown *Bbu* ribosomal proteins is revealed directly through their cryo-EM density: bS22 in the small subunit (Fig. 1c, e) and bL38 in the large subunit (Fig. 1c, f). The uL30 protein in *Bbu* is enlarged through an N-terminal extension (Fig. 1c, g) that occupies the same site as the much smaller helical ribosomal protein bL37 in mycobacterial ribosomes. The secondary structures for the *Bbu* 23S RNA and 5S RNA are depicted in

Supplementary Fig. 2 and the secondary structure for the 16S RNA is depicted in Supplementary Fig. 3. These secondary structure depictions are consistent with the tertiary structure of the ribosomal RNAs modeled into the *Bbu* 70S cryo-EM density.

### The bbHPF protein and its ribosome interactions

The single 97 amino-acid (aa) residue hibernation promoting factor gene in *Bbu*, named *bb0449* in the KEGG database[23] (Uniprot ID: A0A8F9U6W1), was previously reported to have low mRNA transcript and expressed protein levels during various growth phases[24]. The expressed protein did not localize to the ribosome-associated protein fraction, and its gene deletion did not affect the mouse-tick infectious cycle[24]. Based on these observations, it was suggested that the 97 aa-residue bbHPF may not play its conventional role in ribosome hibernation[24]. There are two known mechanisms for formation of a hibernating ribosome by dimerization in bacteria: (a) The ribosome modulation factor (RMF) protein, e.g., in *Escherichia coli* (*Eco*), first helps form a 90S ribosome dimer that then gets converted to a 100S dimer by binding of the shorter form of HPF protein[16], (b) a longer form of HPF protein with a C-terminal dimerization domain, e.g., in *Staphylococcus aureus* (*Sau*), directly mediates the formation of a 100S dimer by forming a dimer itself with a second HPF C-terminal domain[10]. The *Bbu* genome has no annotated RMF or longer HPF protein, so a 70S hibernating ribosome is more probable than a 100S hibernating ribosome dimer.

The bbHPF protein is a shorter HPF resembling the *E. coli* YfiA and HPF proteins and differs from "long" HPFs in lacking a C-terminal dimerization domain, which forms a dimerization interface that stabilizes a hibernating 100S ribosome dimer in certain bacteria[10,12,15,17]. In our structure, we find bbHPF bound to the decoding center in the small subunit of the *Bbu* 70S monosome, thereby confirming the sequence-based expectation that it does play a role in 70S ribosome hibernation (Fig. 1c, h). Its binding site overlaps with the expected binding site for doxycycline suggesting that simultaneous binding of the two may not occur. An E-site tRNA was found interacting with the C-terminal end of bbHPF with no mRNA density near it. Maintenance of the bbHPF and E-site tRNA densities in all sub-classes of the 70S ribosome obtained

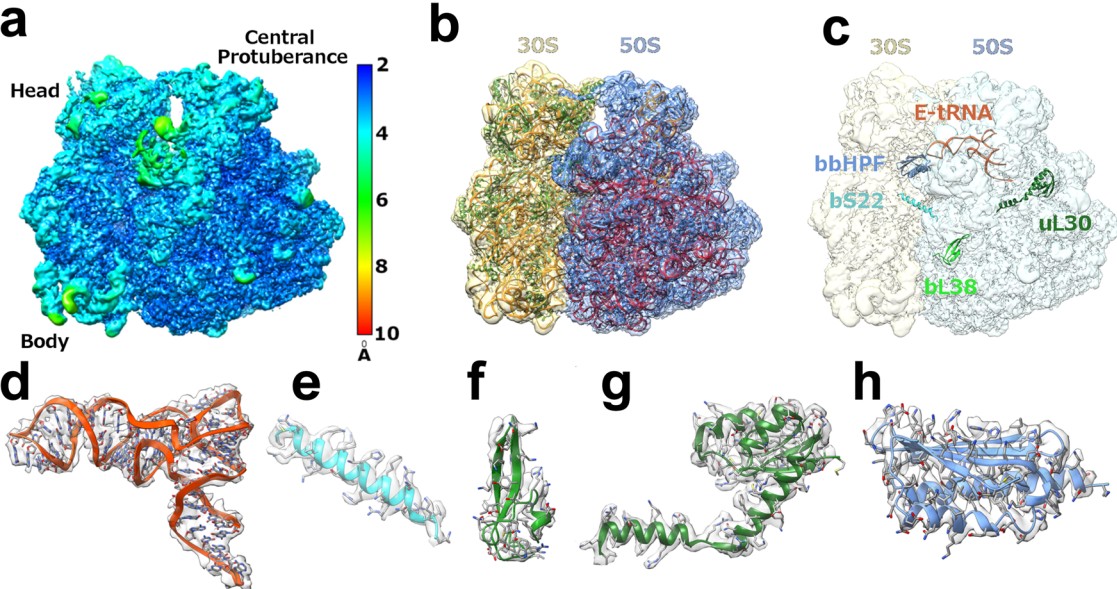

**Fig. 1 | *Bbu* 70S hibernating ribosome structure and protein components.** **a** Cryo-EM density map of 70S with local resolution indicated by color. **b** Model fit into cryo-EM density map with 30S density in khaki, 50S density in blue, 16S RNA and 5S RNA in orange, 23S RNA in red, 30S proteins in green, and 50S proteins in blue. **c** Notable components in the structure shown within the cryo-EM density map. These components of the hibernating 70S ribosome, shown in ribbon format with known sidechains in stick format, are **d** the E-site tRNA (orange), **e** the bS22 protein (cyan), **f** the bL38 protein (green), **g** the longer uL30 protein (dark green), and **h** the bbHPF protein (blue).

**Table 1 | Curated sequence alignment for bacterial HPFs with known structures**

| ID | Aligned sequence |
|---|---|
| | β1 　　　　 α1 　　　　　 β2 　　　　 β3 　　　　 β4 　　　　 α2 |
| | bbb bbb　　aaaaaaaaaaaaa　　aaa　bbbbbbb　　bbbbbbbb　　bbbbbbb　aaaaaaaaaaaaaaaaaaaaaaaaa |
| *Bbu* | ----MEPKIQ-TVNYSLNENEKNFILKKLEKFDTHIKKHIDNLKITIKKEH-----ELFKLDAHIHFN-W-GKIIHIREDGKILLNLIDSAIARLYKTATKEKEKKNNK---- |
| *Sau* | ----IRFEIH-GDNLTITDAIRNYIEEKIGK-LERYFNDVPNAVAHVKVKTYSN--SATKIEVTIPLK---NVTLRAEERNDDLYAGIDLINNKLERQVRKYKTRINRKSRDR |
| *Tth* | ---MNIYKLI-GRNLEITDAIRDYVEKKLAR-LDRYQDGELMAKVVLSLAGSPHVEKKARAEIQVDLP---GGLVRVEEEDADLYAAIDRAVDRLETQVKRFRERRYVGKRHS |
| *Eco1* | ---TMNIT-SKQMEITPAVHVADRLAK-LEKWQTHLINPHIILSKEP-----QGFVADATINTP---NGVLVASGKHEDMYTAINELINKLERQLNKLQHKGEARRAA- |
| *Eco2* | ----MQLNIT-GNNVEITEALREFVTAKFAK-LEQYFDRINQVYVVLKVEK-----VTHTSDATLHVN---GGEIHASAEGQDMYAAIDGLIDKLARQLTKHKDKLKQH---- |
| *Cbu* | ----MHIQMT-GQGVDISPALRELTEKKLHR-IQPCRDEISNIHIIFHINK-----LKKIVDANVKLP---GSTINAQAESDDMYKTVDLLMHKLETQLSKYKAKKGDHR- |
| *Vch* | ----MQINIQ-GHHIDLTDSMQDYVHSKFDK-LERFFDHINHVQVILRVEK-----LRQIAEATLHVN---QAEIHAHADDENMYAAIDSLVDKLVRQLNKHKEKLSSH---- |
| *Msm* | ERPHAEVVVK-GRNVEVPDHFRTYVSEKLSR-LERFDKTIYLFDVELDHERNRRQ-RKNCQHVEITARGR-GPVVRGEACADSFYTAFESAVQKLEGRLRRAKDRRKIHYGDK |
| *Aba* | ----MNIEIRTDKNIHNSERLITYVRAELTQEFQRHSERITHFSVHFSDENGDKG-GDKDIHCMIEARPSGLKPVAVHHKAGNIDASIHGAIEKLKRSLEHTFEKKEHPRGGQ |
| | 　　　　　　:　　　　　　..　:　　　　　　.　　　　　.　　:　　　　　:　　　:　..　　*　　:　 |

Sequences initially aligned using Clustal2.1. *Bbu* - *Borreliella burgdorferi* (O51405), *Sau* - *Staphylococcus aureus* (D2Z097), *Tth* - *Thermus thermophilus* (Q5SIS0), *Eco1* - *Escherichia coli* YfiA (P0AD49), *Eco2* - *Escherichia coli* HPF (P0AFX0), *Cbu* - *Coxiella burnetii* (Q83DI6), *Vch* - *Vibrio cholerae* (H9L4N9), *Msm* - *Mycobacterium smegmatis* (A0QTK6), *Aba* - *Acinetabacter baumannii* (V5V8V8). Uniprot IDs for proteins in parentheses after species name. BbHPF numbered secondary structure elements shown on top as α-helix (**a**) or β-strand (**b**). Bacterial HPF sequences are truncated to within four residues of bbHPF sequence.

through multiple 3D classifications suggest that bbHPF and E-site tRNA colocalize on the hibernating *Bbu* 70S ribosome and are not a combined density due to two separate bbHPF bound or E-site tRNA-bound *Bbu* 70S ribosome populations.

Including the bbHPF structure reported in this work, atomic resolution structures for eight HPF proteins from seven different bacterial species are now known (Supplementary Table 2). When aligned by primary protein sequence, multiple positions in these HPF proteins have similar amino acids, but Leu83 is the only fully conserved amino-acid residue (Table 1 and Supplementary Table 3). This suggests that a certain level of sequence divergence is tolerated in the HPF domain that binds at the decoding center of bacterial ribosomes during their hibernation. This HPF domain is structurally similar in all these bacterial species, having a fold with two α-helices and four β-strands in a β1-α1-β2-β3-β4-α2 topology, with β1 and β2 strands forming a parallel β-sheet and the β2, β3, and β4 strands forming an antiparallel β-sheet continuous with the first parallel β-sheet. There seems to be permissibility for an increase in the size of some of the β strands or α helices (e.g., β1 or α1), a break in the helicity of an α-helix (α1), and increases in the sizes of the loops between β-strands (β2-β3 or β3-β4) (Table 1). The YfiA protein (Uniprot ID: P0AD49) and the HPF protein (Uniprot ID: P0AFX0) in *Eco* differ in sequence but both bind the same decoding center site in the *Eco* ribosome, indicating intraspecies sequence permissibility in HPF-ribosome interaction. There are multiple structures of the *Eco* YfiA bound to the *Thermus thermophilus* (*Tth*) ribosome[16,25–31], indicating that inter-species permissibility also exists in HPF-ribosome interaction. There is also internal structural variability in the HPFs, as seen in the overlay of these known HPF structures aligned to each other (Fig. 2b), and additional positional variability of the HPFs in their binding pocket, as seen in the overlay of these known HPF structures bound to a bacterial ribosome with the structures aligned using the 16S ribosomal RNA (Fig. 2c). Taken together, this evidence suggests that occupation of the bacterial small subunit decoding center binding site does not require a highly conserved HPF sequence, a very rigid HPF structure, or a very restrictive positioning of the HPF in its ribosomal binding site.

Including the structure reported in this work, there are four ribosome structures with HPF proteins and E-site tRNA both modeled into the density map. These include a hibernating *Mycobacterium smegmatis* (*Msm*) ribosome structure (PDB ID: 5ZEP[32]) and two *Eco* hibernating ribosome structures (PDB IDs: 6H4N[11], 6Y69[33]). When aligned using the HPF protein, there is a clear relative motion revealed between the HPF protein and the E-tRNA in the four structures (Supplementary movie 2). The *Msm* structure and the *Eco* structures show

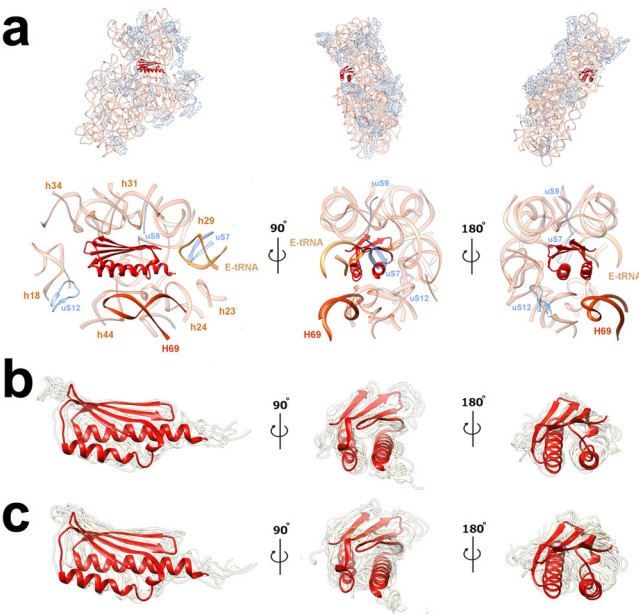

**Fig. 2 | The bbHPF binding site orientation and bacterial HPF protein structural variability. (a)** Subunit interface (left), mRNA exit (middle), and mRNA entry (right) site views of bbHPF in its decoding center binding site in the *Bbu* ribosome with full 30S view shown above. Labels indicate location of ribosomal proteins (in blue) uS7, uS9, uS12, E-site tRNA (in yellowish orange, labeled E-tRNA), and large subunit RNA helix 69 (in dark orange, labeled H69) of the 23S ribosomal RNA. 16S ribosomal RNA shown in semitransparent light orange. Known bacterial HPF structures, with resolutions better than 3.5 Å, shown in transparent khaki, **b** aligned by HPF protein residues, showing internal structural variability, and **c** aligned by 16S ribosomal RNA residues, showing position variability in their ribosomal binding site. BbHPF is shown as opaque red in all panels. Views shown in panels **b, c** are analogous to those shown in panel (**a**). Details of structures used in panels **b, c** are listed in Supplementary Table 2.

substantially different relative HPF and E-tRNA orientations and the *Bbu* structure shows an intermediate relative orientation (Supplementary Fig. 4). When aligned using the small subunit 16S RNA, both the HPF protein and the E-tRNA adjust their relative positions with respect to the ribosome such that overall binding of the two appears roughly similar in all four structures (Supplementary Fig. 5). HPF and the anti-codon stem loop of E-tRNA interact, which likely helps inhibit ribosome activity[11]. The larger difference between the HPF and E-tRNA

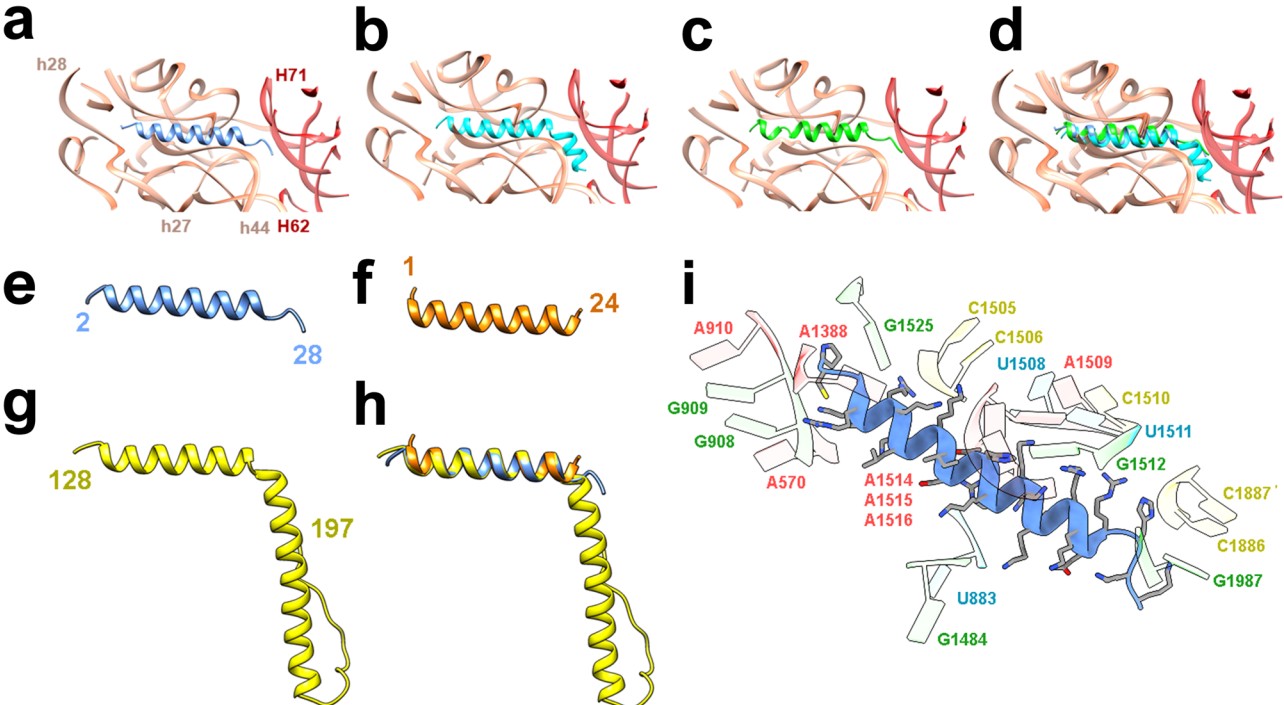

**Fig. 3 | The bS22 protein structure in *Bbu* and other organisms. a** *Bbu* bS22 (blue) in the *Bbu* ribosome binding pocket; **b** *Msm* bS22 (cyan) in the *Msm* ribosome binding pocket; **c** *Fjo* bS22 (green) in the *Fjo* ribosome binding pocket; **d** Overlay of all three bS22 proteins and their respective binding pockets showing the high structural similarity in the three binding pockets with 16S RNA shown in orange and 23S RNA shown in red; **e** *Bbu* bS22 protein; **f** *Homo sapiens* (*Hsa*) eL41 protein; **g** *Hsa* mitochondrial mS38 protein; **h** overlay of *Bbu* bS22, *Hsa* eL41, *Hsa* mS38 proteins; **i** Molecular interactions of *Bbu* bS22 protein with the ribosomal RNA components of the binding pocket, interacting 16S RNA helix numbers in orange, interacting 23S RNA helix numbers in red.

relative orientations for *Msm* and *Eco/Bbu* may originate from the *Msm* HPF being a longer HPF with a C-terminal domain and its neighboring E-tRNA adjusting to the presence of its linker to its C-terminal domain. There is no mRNA density modeled near the E-tRNA in any of these structures, suggesting that the C-terminal end of the HPF protein and the 30S environment in that region help stabilize the anti-codon stem loop part of the E-tRNA in hibernating ribosomes.

**Small ribosomal subunit**

The *Bbu* small subunit unexpectedly showed a clear density for a helical protein in the binding pocket seen to be occupied by the bS22 protein in mycobacterial[14,18,19] and Bacteroidetes[20] ribosome structures. There was no annotated bS22 protein in the *Bbu* genome necessitating identification of the protein sequence through a translated nucleotide genome search. *Borrelia* species can be phylogenetically grouped into three lineages, Lyme borreliae, relapsing fever borreliae, and reptile- and echidna-associated *Borrelia*[34]. Using an approach described in the Supplementary note 1, the bS22 protein sequence (*bbO822*) was identified as using a non-canonical GUG start codon and confirmed in two ways: (a) modeling of the protein sequence into the cryo-EM density map and finding good sidechain density fits (Fig. 1e), (b) finding the same conserved sequence protein in all three groups of *Borrelia* species (Supplementary Table 4). *bbO822* spans the nucleotide range 867605-867697 in the *Bbu* chromosome (Genbank ID: AE000783.1). A transposon library in *Bbu* showed transposon insertions in only 35% of genes in the linear chromosome with the *bbO822* gene encoding bS22 grouped among the 601 essential gene candidates with no insertions[35]. The bS22 transcript is also highly abundant in fed nymphs and its production was not significantly different between infected hosts and tick vectors[36].

The *Bbu* bS22 protein has greater structural and sequence homology with the *Flavobacterium johnsoniae* (*Fjo*) bS22 protein than the *Msm* bS22 protein (Fig. 3, Supplementary Note 1). It is highly basic,

with 61% of its residues being either lysine or arginine. This presumably aids its binding to an RNA pocket formed by the 16S RNA helices h2, h27, h44, h45, and its interaction with the 23S RNA helix H68 (Fig. 3) upon association with the 50S ribosomal subunit. Amongst the three groups of *Borrelia* species, only one residue change is observed in this protein sequence (K10Q), which occurs in *Borrelia miyamotoi*, *Borrelia hermsii*, *Borrelia parkeri*, *Borrelia venezuelensis*, and *Borrelia turicatae*, while all other amino acids are completely conserved (Supplementary Table 4). This protein is likely to be discovered in other bacterial species since its shorter length and non-canonical start codon possibility may pose difficulty for automated protein annotation tools and the ribosomal RNA pocket to which it binds is structurally conserved. Just like *Msm* and *Fjo* bS22 proteins, the *Bbu* bS22 protein is analogous to the eukaryotic mitochondrial small subunit protein mS38[37–39] and the eukaryotic cytosolic protein eL41[40] (Fig. 3e–h). The mS38 protein, also called the Aurora kinase A interacting protein, is a longer protein with a bent helical structure in which the bend is at the same location as in the *Bbu*, *Fjo*, and *Msm* bS22 proteins (Fig. 3g). The eL41 protein, although named as a large ribosomal subunit protein, is localized within the small subunit-binding pocket in eukaryotic ribosomes[40], except for the thermophilic fungus *Chaetomium thermophilum* (*Cth*), where it is also found in a large subunit pocket[41]. It has only one helix with 24 aa residues, is shorter than the *Bbu* bS22 protein, and has no bend in its structure (Fig. 3f). These similarities among bS22 and its eukaryotic analogs suggest that these ribosomal proteins qualify for a universal name, instead of them being named separately as distinct bacterial, mitochondrial, or eukaryotic ribosomal proteins.

The *Bbu* ribosome resembles the *Fjo* ribosome more than the *Msm* ribosome in another aspect – the presence of the bS21 protein. The bS21 protein in the *Fjo* ribosome, along with the proteins bS6 and bS18, are proposed to help form a binding site for sequestering the Anti-Shine-Dalgarno (ASD) sequence for efficient translation of mRNA transcripts that do not contain a Shine-Dalgarno (SD) sequence[20]. This

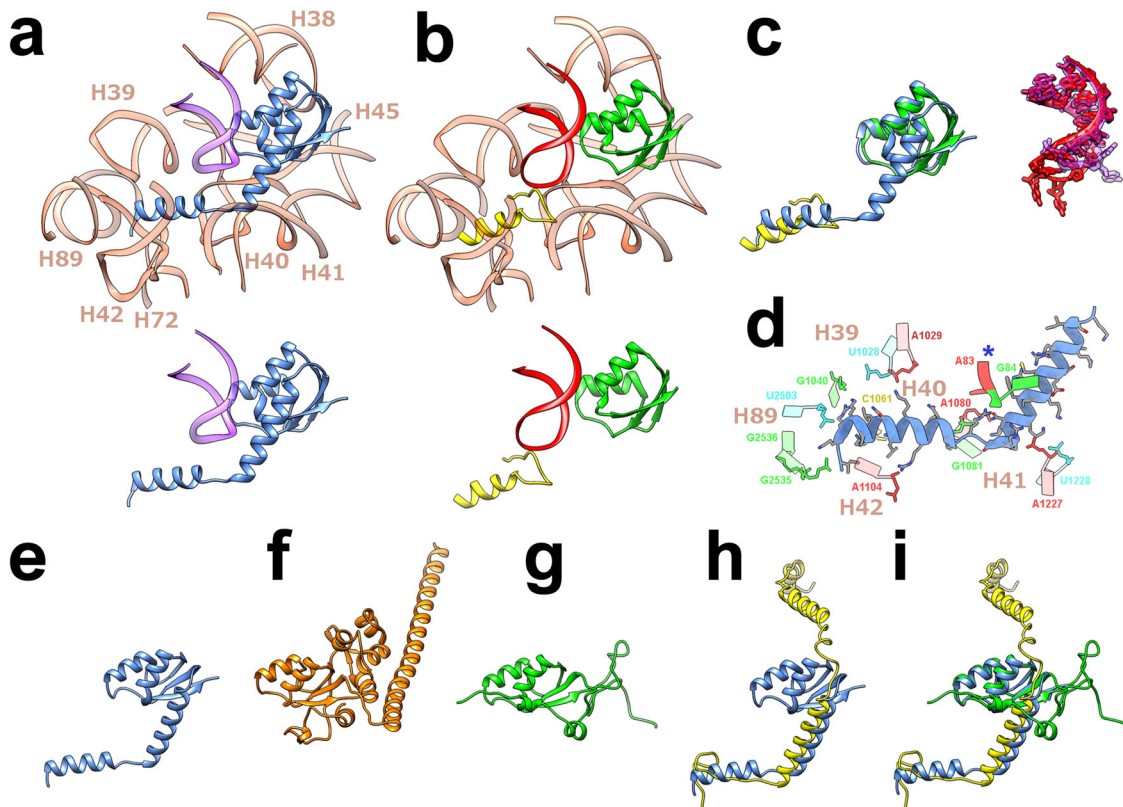

**Fig. 4 | The structure of the longer uL30 protein in *Borreliella* species and its relationship with the bL37 protein in mycobacteria, the *Hsa* uL30 protein, and the *Hsa* mL30 and mL63 proteins. a** The *Bbu* uL30 protein (blue) in its structurally conserved 23S RNA (light orange) and 5S RNA (purple) environment, with the extracted uL30 and 5S RNA shown below for clarity. **b** The *Msm* uL30 protein (green) and the *Msm* bL37 protein (yellow) in its structurally conserved 23S RNA (light orange) and 5S RNA (red) environment, with the extracted uL30, bL37, and 5S RNA shown below. **c** Overlays of the *Bbu* uL30 protein and the *Msm* uL30 and bL37 proteins (left), and the neighboring *Bbu* (purple) and *Msm* (red) 5S RNA segments (right). **d** The ribosomal RNA binding pocket for the uL30 N-terminal extension (residue 2-40), blue asterisk indicates 5S RNA. **e** The *Bbu* uL30 protein. **f** The *Hsa* uL30 protein. **g** The *Hsa* uL30m protein. **h** Overlay of the *Bbu* uL30 protein (blue) and the *Hsa* mL63 protein (yellow). **i** Overlay of the *Bbu* uL30 protein and the *Hsa* uL30m and *Hsa* mL63 proteins.

binding pocket is also present in the *Bbu* ribosome but has some differences (Supplementary Note 2). The bS21 protein's C-terminal end in *Bbu* is longer than that in *Fjo* and forms a well-ordered longer helix (Supplementary Fig. 6). The bS6 C-terminal end is also longer in *Bbu* but like the ASD sequence region of *Bbu* 16S RNA is not resolved in the cryo-EM density map. However, the bS6 C-terminal end does show a propensity to form a short helix like the *Fjo* bS6 in the predicted Alphafold model of full-length bS6 (Supplementary Fig. 4d). Only some of the specific residues implicated in stabilizing the ASD sequestration in *Fjo*[20] are maintained in *Bbu* (*Fjo* bS18 Gln58, Phe50, Leu62, Leu66 are analogous to *Bbu* bS18 Gln59, Phe51, Leu63 and Ile67). Other such residues are altered, e.g., Tyr54 in *Fjo* bS21 is replaced by Lys53 in *Bbu* bS21. Even with these differences, it is possible that this binding pocket in the *Bbu* ribosome can stabilize the ASD sequence in *Bbu* 16S RNA, perhaps to a lesser extent than *Fjo*, which might especially be useful when translating leaderless mRNA transcripts that occur with a high frequency in *Bbu*.

**Large ribosomal subunit**
In the *Bbu* ribosome large subunit, there is a structural feature that has not been previously observed in any bacterial ribosome structure. The uL30 protein in *Bbu* (Fig. 1c, g) is a structural and functional representation of two separate ribosomal proteins – uL30 and bL37 (Fig. 4a, b, Supplementary movie 3). The bL37 protein, a small helical protein that has only been found in mycobacterial ribosomes so far, occupies a pocket near the peptidyl transferase center (PTC). In *Bbu*, there is no separate bL37 protein, but a N-terminal helical extension of uL30

occupies the same pocket by adopting a bent helical structure (Fig. 4a, c). A sequence comparison of the mycobacterial bL37 protein and the *Bbu* uL30 N-terminal extension is shown in Supplementary Note 3. Helical N-terminal extensions of uL30 are present in non-*Borrelia* bacterial species (Supplementary Fig. 7) as well as in other *Borrelia* species (Supplementary Fig. 8), indicating that this feature is not unique to *Bbu*. The binding pocket accommodating the N-terminal *Bbu* uL30 extension is composed of nucleotide residues from the 23S rRNA helices H39-H42 and H89, as well as 5S RNA nucleotide residues (Fig. 4d). The *Bbu* uL30 protein can also likely be expressed in a shorter form not containing the N-terminal helix before the bend (i. e., the helix corresponding to mycobacterial bL37), which suggests that the occupation of the bL37 pocket could be controlled by regulating the relative expression of these shorter and longer forms of *Bbu* uL30 (Supplementary Fig. 8). The *Bbu* uL30 protein resembles the mammalian cytosolic uL30 protein in having an N-terminal helical extension, except that the uL30 N-terminal helical extension in the mammalian cytosolic ribosome is not bent and is oriented in a different direction (Fig. 4f). The *Bbu* uL30 protein also has similarities to mammalian mitochondrial proteins uL30m and mL63, which together occupy the analogous regions in the mammalian mitochondrial ribosomal large subunit (Fig. 4g–i). These observations suggest the possibility that an N-terminal extension of uL30 might have evolutionarily bifurcated into two proteins in mycobacteria and mammalian mitochondria. Ribosomal protein expansions have coevolved in mammalian mitochondrial ribosomes in conjunction with ribosomal RNA reductions[37–39,42–45]. Bifurcation and subsequent expansion of one or

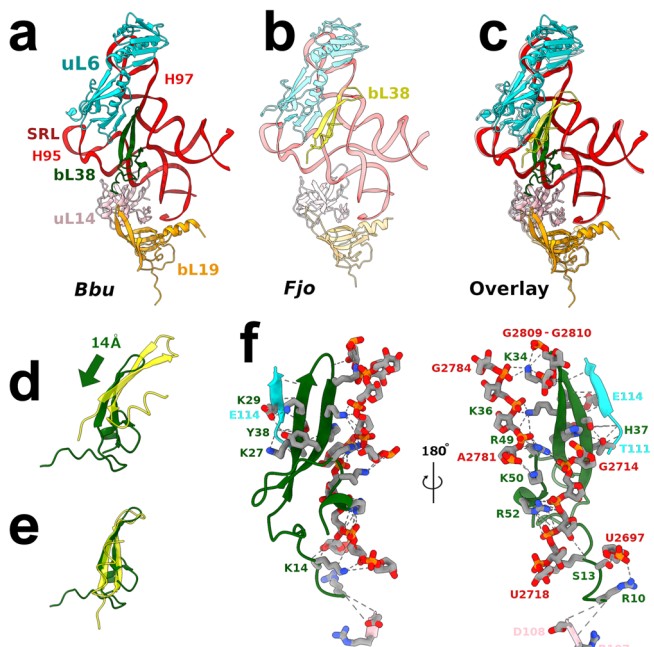

**Fig. 5 | Ribosome interactions of the *Bbu* bL38 protein. a** 50S subunit neighborhood of *Bbu* bL38 (green), **b** 50S subunit neighborhood of *Fjo* bL38 (yellow), **c** Overlay using 23S RNA of **(a)** and **(b)** showing displacement of *Bbu* bL38 as compared to *Fjo* bL38 while the neighborhood remains structurally conserved, **d** relative displacement of *Bbu* bL38 as compared to *Fjo* bL38 in overlay, **e** Internal structural differences between *Bbu* bL38 and *Fjo* bL38 when overlaid on each other, **f** detailed atomic interactions of *Bbu* bL38 with *Bbu* 50S subunit shown from two opposite views. The coloring is as follows: uL6 in cyan, uL14 in pink, bL19 in orange, *Bbu* bL38 in dark green, *Fjo* bL38 in yellow, 23S RNA in red. The *Fjo* models are shown in transparent view to facilitate comparison with *Bbu* in the overlay.

both resulting ribosomal proteins may provide a mechanism by which this evolution occurred.

The *Bbu* ribosomal large subunit also has a clear density corresponding to the recently identified bL38 protein in the *Fjo* ribosome structure (Fig. 1c, f). A ribosomal protein at this location was not expected to be found in *Bbu* as there is no annotated bL38 protein in the *Bbu* genome. We were not able to identify a protein sequence corresponding to this density in the *Bbu* genome through a translated nucleotide search using the *Fjo* bL38 sequence. The density was strong enough to create a reliable backbone model and had some clear sidechain densities that would help distinguish between candidate sequences but our attempts to use de novo methods to identify a sequence for this protein through the cryo-EM density alone were not successful. We first built a well-fitting C-α atom model of a protein based on the cryo-EM density map. We then performed long-gradient Liquid Chromatography Tandem Mass spectrometry (LC−MS/MS) analysis of the proteins in the purified ribosomes. As described in Supplementary Note 4, we were able to use these to unambiguously assign the *Bbu* bL38 protein as the previously unidentified gene *bbO162* (Uniprot ID: O51184).

Although placed in a similar location in the 50S subunit and its binding region being structurally conserved (Fig. 5a–c), the *Bbu* bL38 protein shows some characteristics distinct from the bL38 protein in *Fjo*. The *Bbu* bL38 protein interacts with the 23S RNA helix 95 (H95) containing the α-sarcin-ricin loop (SRL) and may influence the binding of multiple GTPases that interact with the SRL to perform their function (Supplementary Fig. 9). It has 9 residues at its N-terminal end that are not well-resolved in the cryo-EM density. Since *Bbu* bL38 is displaced by 14 Å towards the uL14 and bL19 proteins as compared to the *Fjo* bL38 protein (Fig. 5d), its unresolved

N-terminal residues might be interacting dynamically with these proteins. The *Bbu* bL38 protein also shows an internally distinct structure as compared to *Fjo* bL38 (Fig. 5e), in having a small third β-strand followed by a short α-helix that contain additional positively charged residues participating in a 5-residue (K34, K36, R49, K50, and R52) spine of interaction with the 23S RNA (Fig. 5f). Additional backbone interactions with uL6 and sidechain interactions with residues T111 and E114 from uL6, backbone interactions with uL14, and interactions with multiple residues from the 23S RNA helices H95 and H97 stabilize bL38 binding to the 50S subunit.

## 23S RNA Helix 68 disorder in the large ribosomal subunit

Three-dimensional (3D) classification of the cryo-EM particle images was done to find out if: (a) the subunit ratcheting motion[46] was occurring in the hibernating ribosome; (b) there was a proportion of non-hibernating ribosomes; (c) the HPF protein and the E-tRNA were colocalizing on the 70S ribosome or were a combined density consisting of two different populations; and (d) there were additional protein densities in sub-populations of the particle images obtained. The 3D classification did not yield clearly different structural populations, which suggests that our sample of the hibernating *Bbu* 70S ribosome was mostly homogeneous (Supplementary Fig. 10). Some variability was observed in the presence or absence of parts of the bS2 protein density, but no individual class could be identified with a complete bS2 density present. A small number of particles could be separated into a class mostly corresponding to the 50S subunit with a miniscule proportion of 70S particles within it that could not be eliminated entirely. This class of 12,449 particles was used to reconstruct a *Bbu* 50S subunit structure at a resolution of 3.4 Å (Supplementary Fig. 11a). The protein densities corresponding to the full-length uL30 and bL38 proteins were found unaltered in the 50S subunit structure (Supplementary Fig. 11b–e), confirming that both these proteins are stable components of the 50S subunit.

A surprising finding in this 50S subunit structure was that the long and usually well-resolved 23S RNA helix H68 was not even partially ordered (Supplementary Fig. 12). This disorder, also present for the usually flexible H69, seems to be greater than the alternate conformations identified for H68 in the *Sau* 50S subunit and 70S ribosome at physiological temperature (Supplementary Fig. 13, Supplementary movie 4) that were suggested to be involved in the ribosomal translocation mechanism[47]. A lowering of the density threshold values shows some ordering of H68 but only with simultaneous appearance of the fragmented 30S subunit densities, suggesting that this ordering may be attributable to the small number of 70S particles in this class, which could not be computationally removed. One explanation for the H68 disorder is that the 70S sucrose gradient fraction has some not-fully-assembled 50S population. Known 50S assembly intermediates[48,49] are listed in Supplementary Table 5 and compared to the *Bbu* 50S map in Supplementary Fig. 14. Since no proteins are missing in the *Bbu* 50S structure and the only unusually disordered 23S RNA region is helix H68, it could be a late 50S assembly intermediate structure.

Another explanation is that this 50S population is formed through splitting of a small 70S sub-population. Conformational variability of helices H67-H71 of the 23S rRNA recently observed in cryo-EM structures of log-phase *Msm* ribosomes was proposed to prevent association of the 50S subunit with the 30S subunit[50]. H68 having a larger conformational variability in *Bbu* would also be incompatible with proper 70S structure maintenance. The nearby inter-subunit bridge B7a near the L1 stalk region may be one of the final ones to dissociate during ribosomal subunit splitting[51] and larger-scale H68 motions could affect this bridge. With such larger motions, the H68 helix could therefore play a role in completing the dissociation of the two *Bbu* ribosomal subunits. If so, the lack of any special features in H68 in *Bbu*

as compared to other bacterial species suggests that this postulated ribosome splitting assistive role for H68 might be common to other ribosomes as well.

## Antibiotic-binding pockets

This *Bbu* 70S ribosome structure captures details of antibiotic-binding pockets that allow construction of detailed models of antibiotics bound to it. We have modeled the structure of three antibiotics, two that are in clinical use already (doxycycline and erythromycin), and one that has recently been shown to be of great promise in treating Lyme infections (hygromycin A[22]) using de novo docking (Supplementary Fig. 15) and structural analogy to previous antibiotic-bound ribosome structures (Supplementary Table 6, Supplementary Fig. 16). The de novo docking, done in the vicinity of the expected binding sites, found binding poses analogous to the previous experimental structures with the Autodock Vina[52] estimated binding free energies of −5.4 kcal/mol for doxycycline, −6.2 kcal/mol for erythromycin, and −7.3 kcal/mol for hygromycin A. A comparison between the hygromycin A binding site for *Bbu* and the same site in the *Tth* ribosome, with[53] and without[25] hygromycin A bound, reveals that some 23S RNA residues in *Bbu* are intrinsically positioned in conformations forming a more open binding pocket conducive to hygromycin A binding (Supplementary Fig. 17). Recent studies have shown that antibiotics may perform their function by introducing protein synthesis malfunctions in a context-specific manner[54,55]. In their empty ribosome binding modes, these antibiotics sterically overlap with components that bind to the translating ribosomes suggesting that they may adjust their ribosome binding modes to the presence of these other components. Structurally characterizing *Bbu* translation factors and *Bbu* tRNAs bound to its ribosome in various steps of protein translation, in the presence and absence of antibiotics, can help clarify their context-specific mechanism of action.

In summary, this study identifies unanticipated alterations in the *Bbu* ribosome, such as the presence of bS22, bL38, and an N-terminal extended uL30 also assuming the role of the bL37 protein. The extended uL30 being present in ancient bacterial species (Supplementary Fig. 7) as well as having eukaryotic cytosolic and mitochondrial ribosome analogies (Fig. 4e–i) suggests this to be analogous to an earlier version of uL30 in evolution that has subsequently shortened or split or expanded, which suggests an underlying mechanism for the increase in ribosomal protein numbers and masses in eukaryotic mitochondrial and cytosolic ribosomes. Our hibernating *Bbu* 70S ribosome structure thus provides the groundwork for better understanding ribosome dormancy, ribosome evolution, antibiotic mechanisms of action, and for development of new structure-based antibiotic therapeutics for Lyme disease.

## Methods

### Ribosome purification

The *Bbu* strain B31-A3[56] was inoculated in 2L BSK II complete medium and grown to the late logarithmic phase as we were interested in hibernating ribosomes, which have the additional advantage of being available in relatively large numbers in a smaller volume of cell culture. Cells were harvested and lysed by French press for two pressings at 16,000 PSI and the lysate centrifuged twice at $11,952 \times g$ for 30 min. Formation of a pellet of spirochete cells, expected to occur after spinning at $7649 \times g$, was observed after the first spin. The second spin yielded a much smaller pellet. The supernatant after the second spin was examined by dark microscopy at 400X magnification for intact spirochetes. After observing cell debris from approximately three dead spirochetes in the lysate, a final spin of $34,541 \times g$ for 45 min was done, which ensured pelleting of all the spirochete cell debris. The rest of the ribosome purification protocol[57] is as follows. The supernatant was collected in a Beckman PC ultracentrifuge tube and centrifuged for 2 h and 15 min at $188,043 \times g$ in a Beckman rotor Type 70Ti. Pellets

were soaked in 2 mL of HMA-10 buffer (20 mM HEPES K pH 7.5, 600 mM NH4Cl, 10 mM MgCl2, 5 mM β-mercaptoethanol) and kept in an ice bath overnight. 16 mL of HMA-10 buffer was added and the preparation was put on a shaking rocker for one hour at 4 °C with 3 units/mL Rnase-free Dnase (Ambion) added. The contents were transferred to Beckman PA tubes and centrifuged at $21,986 \times g$ at 4 °C for 15 min in Beckman rotor JA 30.5Ti. The supernatant was collected in Beckman ultracentrifuge PC tubes and centrifuged for 2 h and 15 min at 4 °C at $185,416 \times g$ in a Beckman rotor Type 70Ti. Pellets were resuspended in 200 μL HMA-10 buffer and kept in an ice bath in a cold room at 4 °C overnight. The sample was then centrifuged for 6 min at $17,348 \times g$. The supernatant containing the ribosomes was collected and quantified by measuring optical density at 260 nm. This crude ribosome preparation was then layered on top of sucrose density gradients (10%–40%), prepared in 11 mL tubes containing HMA-10 buffer for 17 h at $55,408 \times g$ in a Beckman rotor SW 41Ti. Ribosome fractions containing primarily 70S monosomes were collected after fractionating the sucrose gradient in a 260 nm Teledyne ISCO gradient analyzer. These pooled 70S fractions were pelleted by ultracentrifugation at $188,043 \times g$ for 6 h in a Beckman rotor Type 70Ti, suspended in HMA-10 buffer, and quantified by measuring absorbance at 260 nm.

### Protein mass spectrometry

The proteins in two 11 μL aliquots of a 350 nM *Bbu* ribosome preparation were reduced and alkylated to remove disulfide linkages, digested using trypsin, desalted, and then examined by long-gradient Liquid Chromatography tandem mass spectrometry (LC–MS/MS), using a Thermo Scientific LTQ-Orbitrap XL mass spectrometer. The protein sequences were identified by comparison with a *Bbu* reference proteome (1290 sequences) downloaded from Uniprot in 2019 (https://www.uniprot.org/proteomes?query=BORBU). The results of this analysis, performed using the Scaffold5 software, are shown in Supplementary Table 7. The bL38 protein that was not identifiable using other means, such as de novo sequence prediction from the cryo-EM density or a translated nucleotide search using the *Fjo* bL38 sequence, was identified successfully by this analysis. In addition, the bbHPF protein and all other ribosomal proteins except bS22 were also identified at 100% protein identification probability.

### Grid preparation and imaging for cryo-EM

A home-made thin carbon film was coated as a continuous layer (about 50 Å thick) onto Quantifoil 300-mesh 1.2/1.3 grids. These grids were then glow-discharged for 30 s on a plasma sterilizer to hydrophilize the carbon film. The purified ribosome sample (4 μL) was transferred onto each grid after mounting it on a Thermofisher Vitrobot Mark IV system and the grid was maintained for 15 s at 4 °C and 100% humidity. Each grid was then blotted for 5 s with a force offset of +2 and then plunged into liquid ethane. Movies were collected in counting mode using Leginon software[58] on a Thermofisher Titan Krios G3 electron microscope operating at 300 kV with a Gatan BioQuantum imaging energy filter and a Gatan K2 direct electron detection (DED) camera. The cryo-EM data collection details are shown in Table 2. A total of 4661 movies, each having 50 frames, each frame collected every 0.2 s, were obtained at a physical pixel size of 1.0961 Å. Manual curation of the movies using parameters such as estimated ice thickness and contrast transfer function (CTF) fit criteria yielded 2331 movies that were used for further cryo-EM image processing. An electron dose rate of about 8.1 electrons/pixel/second and an exposure time of 10 s yielded a total dose of 67.5 electrons/Å$^2$.

### Data processing for cryo-EM

Movies were processed using patch motion and patch contrast transfer function (CTF) correction implemented in cryoSPARC v2.15[59]. The micrographs were curated to remove those with CTF fits worse than

**Table 2 | Cryo-EM data collection and model refinement details**

| Parameters | 70S ribosome | 50S ribosomal subunit |
|---|---|---|
| **Cryo-EM** | | |
| Electron microscope | Titan Krios G3 | Titan Krios G3 |
| Voltage (kV) | 300 | 300 |
| Pixel size (Å) | 1.0961 | 1.0961 |
| Defocus range (µM) | 0.8–2.5 | 0.8–2.5 |
| Electron dose/image | 67.5 | 67.5 |
| Particles | 288,776 | 12,449 |
| Software | cryoSPARC v2.15 | cryoSPARC v2.15 |
| Symmetry | C1 | C1 |
| FSC threshold | 0.143 | 0.143 |
| Average B-factor (Å2) | 57.2 | 96.6 |
| Resolution (Å) | 2.9 | 3.4 |
| **Atomic model** | | |
| R.M.S. deviations | | |
| Bond lengths (Å) | 0.000 | 0.000 |
| Bond angles (°) | 0.089 | 0.103 |
| Resolved protein residues | 6239 | 3569 |
| Protein rotamer outliers | 22 (0.4%) | 13 (0.4%) |
| Ramachandran favored | 5831 (93.5 %) | 3280 (93.0 %) |
| Ramachandran outliers | 1 (0.016%) | 1 (0.028%) |
| Resolved RNA residues | 4646 | 2867 |
| RNA sugar pucker outliers | 64 (1.4%) | 33 (1.2%) |
| RNA suite outliers | 790 (17.0%) | 428 (14.9%) |
| RNA Suiteness | 0.61 | 0.64 |
| Clash score | 7.5 | 7.1 |

5 Å and those with relative ice thickness greater than 1.05. Template-based auto-picking of particles was done using the template picker module in cryoSPARC with the input 2D templates (100 equally spaced views) obtained from an earlier 3.9 Å resolution reconstruction volume of the hibernating *Bbu* ribosome generated from data collected on the local JEOL3200FSC 300 KV electron microscope using our automation protocol for SerialEM[60]. The automated particle picks were filtered for particles with normalized correlation coefficient (NCC) > 0.2 and a signal amplitude between 3175 and 5454, which selected a total of 541,319 particles and excluded 98,553 particles. The selected particles were extracted from the corrected micrographs at a box size of 380 pixels and then further filtered using multiple iterations of 2D classifications, each followed by selection of good particle classes prior to using them as input for the next 2D classification, yielding a final selection of 288,776 particles used for further processing. These particles were used for generating a 3D reconstruction using ab initio reconstruction, homogeneous refinement, non-uniform refinement[61], and finally local non-uniform refinement with the mask from the previous refinement, whose final gold-standard 3D refinement resolution was 2.9 Å. Further classification was attempted using 3D variability analysis, heterogeneous refinement, and ab initio reconstruction with multiple classes, but the resulting classes showed only small differences from each other, such as presence or absence of fragmented partial density of uS2 protein, suggesting that the particles mostly represented a homogeneous population. The only exception was a small class of 12,449 particles consisting of mostly 50S subunit particles with a slight contamination of 70S particles, whose final gold-standard resolution was 3.4 Å.

### Model building and refinement
The initial models for the 23S, 16S, and 5S ribosomal RNAs were built by manually aligning the *Bbu* sequences to the corresponding sequences of the mycobacterial[18] ribosomal RNAs and using RNA homology modeling program Moderna[62] to generate initial homology models using a known RNA structure (PDB ID: 5O61)[18]. The RNA secondary structures for the *Bbu* 23S, 16S, and 5S ribosomal RNAs were obtained from their modeled 3D structures using the RNA PDBEE server[63,64] and, with the aid of the R2DT server[65], were manually converted to the image format provided in the supplementary material. Initial atomic models for all *Bbu* ribosomal proteins and bbHPF were built using Alphafold2[66]. Manual rebuilding of incorrectly modeled regions was performed using UCSF Chimera v1.16[67] followed by restrained local real-space refinement in Phenix v1.18[68].

The UCSF ChimeraX[69,70] matchmaker module was used for structural overlays of proteins (command line: mm #1 to #0) or RNA (command line: mm #1 to #0 matrix Nucleic). The predicted Local Distance Difference Test (pLDDT) score statistics for starting Alphafold models for proteins included in the reported structures are shown in Supplementary Table 8. The details of RNA structure overlays for antibiotic-bound structures are shown in Supplementary Table 9 and for HPF-bound structures are shown in Supplementary Table 10. The antibiotic docking was performed using Quickvina[71] with the Autodock Vina scoring function[52]. The structures for the antibiotics and ribosome were prepared using the Autodock[72] prepare_ligand4.py and prepare_receptor4.py programs. Initial structures for the antibiotics came from their highest resolution ribosome-bound conformation. The hygromycin sugar ring conformations, which started from their experimentally known conformation, were not expected to vary during docking. The docking was performed with a box centered around the expected antibiotic-binding site, with the exhaustiveness parameter set to 32 and number of modes set to 100. These parameters yielded correct docked poses for hygromycin A in its binding site in the *Thermus thermophilus* (*Tth*) ribosome (PDB ID: 5DOY[53]) with a predicted binding free energy of −9.0 kcal/mol, tetracycline in its binding site in the *Eco* ribosome (PDB ID: 5J5B[73]) with a predicted binding free energy of −5.7 kcal/mol, and erythromycin in its binding site in the *Sau* ribosome (PDB ID: 6SOZ[74]) with a predicted binding free energy of −5.9 kcal/mol. UCSF Chimera v1.16[67] or UCSF ChimeraX version 1.4[69,70] were used to make the molecular figures, Inkscape was used to edit the RNA secondary structure depictions, and Gimp v2.10 was used to make composite figures.

### Reporting summary
Further information on research design is available in the Nature Portfolio Reporting Summary linked to this article.

### Data availability
Cryo-EM volumes and atomic models have been deposited at the EMDB and PDB databanks, respectively. The PDB entries are 8FMW (*Bbu* 70S hibernating ribosome structure) and 8FN2 (*Bbu* 50S subunit structure). The EMDB entries are EMD-29298 (Bbu 70S hibernating ribosome structure) and EMD-29304 (Bbu 50S subunit structure). Source data are provided with this paper.

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

## Acknowledgements

We acknowledge the use of the Wadsworth Center cryo-EM facility and the help and training provided by Chyongere Hsieh and Michael Marko for this use. We thank Joseph Wade for discussions that helped identify the bS22 protein sequence. We thank Caleb Mallery, whose graduate rotation project of antibiotic design in ribosome structures informed the antibiotic docking and structural analysis reported in this study. The cryo-EM data was collected at the Simons Electron Microscopy Center and National Resource for Automated Molecular Microscopy located at the New York Structural Biology Center, supported by grants from the Simons Foundation (349247), NYSTAR, and the NIH National Institute of General Medical Sciences (GM103310). The mass spectrometric analysis was performed by the MS & Proteomics Resource at Yale University, which is funded in part by the Yale School of Medicine and by the Office of The Director, National Institutes of Health (S10OD02365101A1, S10OD019967, and S10OD018034). This work was supported by the NIH NIGMS grant (GM061576) to R.K.A. R.K.A. also acknowledges support to his lab through NIH R01 grants AI132422, GM139277, and AI155473.

## Author contributions

M.R.S., Y-P.L., R.K.A., and N.K.B. designed the project. A.L.M. grew the Bbu culture, lysed the cells, and ensured spirochete cellular disintegration using light microscopy. P.K. purified the Bbu ribosomes. S.M. and N.K.B. performed the protein mass spectrometric data analysis. R.K.K. and N.K.B. performed the initial electron microscopy at Wadsworth Center. N.K.B. processed the cryo-EM data to generate final 3D volumes. M.R.S., E.K.A., and N.K.B. built protein models. S.R.M. and N.K.B. built ribosomal RNA and tRNA models. N.K.B. generated and refined the final combined atomic model built into the 3D volumes and performed the initial structural analysis. M.R.S., S.R.M., R.K.K., S.M., R.K.A., and N.K.B. contributed to further structural analysis and all authors participated in writing or editing the manuscript.

## Competing interests

The authors declare no competing interests.
