## [Peer Review File · Nature Communications]

The structure of a hibernating ribosome in a Lyme disease pathogenREVIEWER COMMENTS

Reviewer #1 (Remarks to the Author):

Banavali and coworkers concisely conducted a scientific study related to the spirochete bacterial pathogen *Borrelia* (*Borrelia*) *burgdorferi* (Bbu) and its impact on Lyme disease. Please note that I am not an expert in the Cryo-EM method. My work focuses on molecular recognition of protein/nucleic acid-ligand recognition and based on my expertise in computational structural biology and drug design, I have reservations on the modelling approach reported in this work. I have provided the breakdown of the key points below.

Prevalence and impact: Bbu affects more than 10% of the world population and causes Lyme disease in approximately half a million people in the US annually.

Therapy: Antibiotics are used for treating Lyme disease, specifically targeting the Bbu ribosome.

Research method: The researchers used single-particle cryo-electron microscopy (cryo-EM) to determine the structure of the Bbu 70S ribosome at a resolution of 2.9 Å.

Novel findings: The Cryo-EM structure reveals unique characteristics of the Bbu ribosome. Contrary to a previous study, the presence of bbHPF (hibernation promoting factor protein) binding to the decoding center of the small ribosomal 30S subunit is confirmed.

Uncommon ribosomal proteins: The 30S subunit contains a non-annotated ribosomal protein called bS22, which has only been found in mycobacteria and Bacteroidetes so far. The large 50S ribosomal subunit of Bbu contains a recently discovered protein called bL38, also present in Bacteroidetes. Additionally, the protein bL37, previously seen only in mycobacterial ribosomes, is replaced by an N-terminal α -helical extension of uL30.

Evolutionary implications: The presence of uL30 and bL37 proteins, which may have evolved from a longer uL30 protein, suggests an evolutionary relationship. The longer uL30 protein interacts with both the 23S rRNA and the 5S rRNA, potentially providing greater stability to the peptidyl transferase center (PTC) region. This finding draws an analogy to similar proteins found in mammalian mitochondrial ribosomes, hinting at a possible evolutionary pathway for protein expansion in these ribosomes.

Antibiotic design: The study predicts free energies of binding for three antibiotics that target the decoding center or PTC of the Bbu ribosome, which are clinically used for treating Lyme disease. The information gained from the ribosome structure can assist in the design of ribosome-targeted antibiotics for more effective Lyme disease treatment.

This work addresses an important problem and therefore deserves publication (key points above) about addressing major concerns on the modelling. I have following reservations on the modelling approach:

Reservation: Authors should provide details of docking and superimposition in the supporting information.

1. While authors have provided PDB codes in the tables, it is not clear how structural superimposition has been carried out using Chimera or ChimeraX. What was the RMSD? Does Chimera allow superimposition of nucleic acid structures including proteins?
2. Figure S11. authors mentioned about QuickVina and Vina scores. The approach is only clear after reading the caption. Can they please provide a method section? The Cryo-Em is an apo structure, is induced fit or allostery possible in Bbu ribosome?
3. How did the authors prepare the structures of ligands/antibiotics and ribosome for docking? Hygromycin has sugars, how were the ring conformations handled during docking?
4. Authors compared the binding of antibiotics in Bbu ribosome with other published ribosomal-antibiotic structures (table S5). Have they validated the docking protocol on these published structures? What is the rationale for using Quickvina program and Vina scoring?
5. Authors used Alphafold2 models for modelling proteins. What is the confidence score of these models?

Reviewer #2 (Remarks to the Author):

Sharma et al. present the cryo-EM structure of a hibernating ribosome isolated from the bacterial pathogen *Borrelia burgdorferi*, which causes Lyme disease. The authors isolated the ribosome from late log-phase cells for structure determination. The cryo-EM structure resolves the bbHPF hibernating factor bound to the small ribosomal subunit and interacting with an E-site tRNA, which was not previously known for its ability to bind to the ribosome. The structure visualizes an expansion in uL30, which is replaced by bL37 in other ribosomes, and density for a putative ribosomal protein bL38. Finally, the authors conducted computational modeling of antibiotics bound to the active site of the ribosome.

Overall, the cryo-EM work is of good quality and the structure presents some new aspects of a ribosome from a disease-causing pathogen. I have some comments on the current manuscript outlined below:

- The authors could explain why they chose to isolate the ribosome from late log-phase cells and how no 100S ribosomes were observed. Perhaps a shorter cruder purification could help address this issue. For example, what is the purpose of the overnight incubations described in the methods, could this be shortened as it may lead to loss of factors/interactions?
- The interaction between bbHPF and the E-site tRNA is interesting. As the authors did a comparison with previous structure, it is not clear if it carries any functional implications? Is it known if it blocks the tRNA on the ribosome? Perhaps this could be commented on the discussion.
- It would be best if the authors highlight any sequence similarities between the extension of uL30 and bL37 in the shown alignment? This is not clear here how similar or different the sequences are. Additionally, the authors could indicate whether the genes of the two proteins are located close to each other, perhaps on the same operon.
- The authors should explain the rationale for doing a 3D classification and whether any masking was used. This could be explained more in the text, results or methods.
- The observation regarding H68 is interesting and suggests that the classified large subunit obtained for a sub-population of particles could possibly be an assembly intermediate. A conformational change in h68 was previously observed (i.e. PMC5041480, PMC5145266). The authors could compare these structures as a supplementary to draw conclusions/connections to ribosome assembly. Did the authors check for protein content for these classes? (i.e. any lacking EM-densities?)
- It is probably still possible to identify the putative density for bL38 from the cryo-EM structure. Did the authors check for matching secondary structures over PDBeFold, sequence is less relevant here and the protein hits would be based on the fold of the protein. Another possible solution is to perform mass spectrometry on the isolated ribosomes and to try and identify the protein from protein hits.
- For the EM validation, it is best to report bond angles and length to the third decimal and indicate the units in the table. Plots for model vs map and FSC plots and a scheme of the processing should also be indicated together with more information on the data processing process (i.e., total number of micrographs, starting number of particles, etc.).

- The computational work on modeling antibiotics is informative but it would be much more exciting to have visualized the structure with bound antibiotics and explore differences across species.

Reviewer #3 (Remarks to the Author):

The authors present results of Cry-EM analyses of the *Borrelia burgdorferi* hibernating ribosome. Highly detailed structural information is provided. Importantly, the authors also identified several points where the *B. burgdorferi* ribosome differs from those of other Eubacteria, including unique subunits and variant subunits. Insights are also provided on binding sites of therapeutically-important antibiotics. This is a valuable report that will have significant impacts upon the Lyme disease and spirochete fields.

I have only a few comments on the manuscript, which I feel will improve the ability of readers to comprehend this report:

- 1) Lines 32-34. The sentence beginning with "The protein bL37" is confusing, especially since *B. burgdorferi* does not produce a bL37 protein. A clearer sentence: "Protein uL30 contains a unique N-terminal alpha-helical extension, which is similar to the bL37 protein that is known only in mycobacterial ribosomes, suggesting that bacterial ribosome proteins bL37 and uL30 may have evolved from a single, longer protein".
- 2) Lines 47-70. This single paragraph contains three distinct trains of thought. I suggest that it be split into three separate paragraphs, on current line 57 (between "amount each year. and "The ribosome is"), and current line 64 (between "smaller ribosome proteins." and "When diagnosed early").
- 3) Lines 153-154. At this location, please provide an ORF number for the newly-identified bS22 protein. The ORF nomenclature of *B. burgdorferi* type strain B31 will be appropriate.
- 4) Line 158. Please define "Fjo" at this point.
- 5) Lines 162-163. Please state which group/species have the K100 residue change. Although that is in a supplemental table, it would be helpful if this useful piece of information is easily provided to readers, rather than expecting us to go to (and search) another file.
- 6) Line 173. The text says "except for one" - please give us the name of that exception.

Response to Reviewers.

We thank all the reviewers for their thoughtful and constructive comments. A point-by-point response to their comments is given below. The page, paragraphs, and lines with the changes in our manuscript document file are mentioned in our responses.

Reviewer 1

1. While authors have provided PDB codes in the tables, it is not clear how structural superimposition has been carried out using Chimera or ChimeraX. What was the RMSD? Does Chimera allow superimposition of nucleic acid structures including proteins?

We used the ChimeraX matchmaker module to carry out the structural superimpositions, but we believe there are no significant differences in the matchmaker modules between Chimera and ChimeraX. The ChimeraX documentation summarizes the method as creating a pairwise sequence alignment between two structures with residue types and/or secondary structure information used to align the sequences, then fitting the aligned residue pairs. Fitting uses one point per residue: CA atoms in proteins and C4' or P atoms in nucleic acid residues. For proteins, the procedure involves calculation of secondary structure assignments with ksdssp, sequence alignments using the Needleman-Wunsch algorithm with the BLOSUM-62 residue similarity matrix (weight 0.7) and secondary structure scoring (weight 0.3), keeping the sequence alignment for the best-scoring pair of chains, one from the reference model and one from the model to be aligned, and using that alignment to iteratively fit the structures with a cutoff of 2.0 Å. For RNA, the default 'BLOSUM-62' matrix specification is to be replaced by a 'Nucleic' matrix specification. The RMSD values for all superimpositions involving antibiotics and HPF-bound structures have been added as Tables S9 and S10, respectively, in the supplementary material. They are generally small, as the overall structures of the equivalent pruned protein CA atoms or RNA C4'/P atoms being superimposed are very similar across species. Neither Chimera nor ChimeraX allows for superimposition of both nucleic acids and proteins together using Matchmaker. We use the Chimera command line to use matchmaker as follows:

For RNA, model #0 as reference, model #1 aligned to model #0 using matchmaker:

```
mm #1 to #0 matrix Nucleic
```

For protein, model #0 as reference, model #1 aligned to model #0 using matchmaker, uses BLOSUM-62 matrix by default

```
mm #0 to #1
```

A condensed version of this description has been added to the Methods section (page 14, paragraph 2, lines 1-3).

2. Figure S11. authors mentioned about QuickVina and Vina scores. The approach is only clear after reading the caption. Can they please provide a method section? The Cryo-Em is an apo structure, is induced fit or allostery possible in *Bbu* ribosome?

We have added Quickvina docking into the methods section (page 14, paragraph 1, lines 6-19). As the reviewer correctly indicates, it does not include motions of the ribosome in conjunction with motions of

the antibiotic being docked. The decoding center and the peptidyl transferase center (PTC) are very conserved among bacterial ribosomes but induced fit and/or allostery in these antibiotic binding sites cannot be ruled out for the *Borrelia burgdorferi* (*Bbu*) ribosome. To a certain extent, using flexible docking or molecular dynamics simulations with fewer restraints can model such motions, but solving high-resolution structures of the antibiotics bound to the *Bbu* ribosome, which is outside the scope of the present study, would provide a more robust answer.

3. How did the authors prepare the structures of ligands/antibiotics and ribosome for docking? Hygromycin has sugars, how were the ring conformations handled during docking?

The structures for the antibiotics and ribosome were prepared using the Autodock mgltools `prepare_ligand4.py` and `prepare_receptor4.py` programs. Initial structures for the antibiotics came from their highest resolution ribosome-bound conformation. There were no input parameters or carbohydrate-specific scoring functions such as Vina-Carb used. The hygromycin sugar ring conformations, which started from their experimentally known conformation, are not expected to vary during docking. This information has also been added to the Methods section (page 14, paragraph 2, lines 6-19).

4. Authors compared the binding of antibiotics in *Bbu* ribosome with other published ribosomal-antibiotic structures (table S5). Have they validated the docking protocol on these published structures? What is the rationale for using Quickvina program and Vina scoring?

The docking protocol has been tried on three original structures (PDB IDs: 5DOY, 5J5B, 6S0Z), the highest resolution one available, for hygromycin A, tetracycline, and erythromycin, respectively. We have added this information to the methods section of the manuscript (page 14, paragraph 2, lines 14-19). The rationale for using the Quickvina program and Vina scoring was their simplicity of use, their reported accuracy, and their widespread use for such applications.

5. Authors used AlphaFold2 models for modelling proteins. What is the confidence score of these models?

AlphaFold2 uses a predicted local distance difference test (pLDDT) confidence estimate per residue between 0 – 100, with 100 indicating the highest confidence. Since it is a per-residue measure and a predicted structure usually has a mix of high-confidence and low-confidence residues, the range of pLDDT values per structure is large and a single average score per structure is not reported by AlphaFold2. To address this comment, however, we have calculated the average pLDDT, maximum pLDDT, and minimum pLDDT for each AlphaFold2 structure included in the manuscript and added a table to the supporting information reporting this information (Table S8).

Reviewer 2.

The authors could explain why they chose to isolate the ribosome from late log-phase cells and how no 100S ribosomes were observed. Perhaps a shorter cruder purification could help address this issue. For example, what is the purpose of the overnight incubations described in the methods, could this be shortened as it may lead to loss of factors/interactions?

We have previously studied ribosome hibernation in mycobacteria and our specific interest in hibernating ribosome structures was a reason for using late log-phase cells, which had the additional advantage of being available in greater numbers in a smaller volume culture. We have added a sentence to the methods in this regard (page 11, paragraph 3, lines 2-4). The protocol followed for ribosome purification was based on the previous protocol used in our lab for isolation of *Escherichia coli* (*Eco*) or *Mycobacterium smegmatis* (*Msm*) ribosomes. Limiting the overnight incubation, reducing the purification steps, or using tagging and affinity-based purification are all promising future efforts to get active *Bbu* ribosomes from log-phase cells.

We are familiar with two mechanisms for formation of 100S ribosome dimers in bacteria: (a) The ribosome modulation factor (RMF) protein, e.g. in *Eco*, first helps form a 90S ribosome dimer that then gets converted to a 100S dimer by binding of the hibernation promoting factor (HPF) protein; (b) a longer HPF protein with a C-terminal dimerization domain, e.g. in *Staphylococcus aureus* (*Sau*), directly mediates the formation of a 100S dimer by forming a dimer itself with a second HPF C-terminal domain. The *Bbu* genome has no annotated RMF or longer HPF protein and therefore may not have a mechanism to form 100S ribosome dimers. Mycobacteria also do not have 100S dimer formation reported in ribosome hibernation. We have added a few lines to the manuscript in this regard (page 4, paragraph 3, lines 7-15).

The interaction between bbHPF and the E-site tRNA is interesting. As the authors did a comparison with previous structure, it is not clear if it carries any functional implications? Is it known if it blocks the tRNA on the ribosome? Perhaps this could be commented on the discussion.

A previous study reporting a structure (PDB ID: 6H4N) with the E-site tRNA and HPF bound together has implied that the HPF and the anticodon stem loop of E-tRNA interact to stabilize each other and inhibit ribosome activity (Beckert et al. Structure of a hibernating 100S ribosome reveals an inactive conformation of the ribosomal protein S1. Nat Microbiol 3, 1115–1121, 2018, <https://doi.org/10.1038/s41564-018-0237-0>). The larger difference between the HPF and E-tRNA relative orientations for *Msm* and *Eco/Bbu* may originate from the *Msm* HPF being a longer HPF with a C-terminal domain and its neighboring E-tRNA adjusting to the presence of its long linker to its C-terminal domain. These points have been added to the manuscript (page 5, paragraph 2, lines 12-17).

It would be best if the authors highlight any sequence similarities between the extension of uL30 and bL37 in the shown alignment? This is not clear here how similar or different the sequences are. Additionally, the authors could indicate whether the genes of the two proteins are located close to each other, perhaps on the same operon.

The *Msm* bL37 and *Bbu* uL30 N-terminal extension sequences align with a sequence identity of 21% :

Msm bL37 MAKRGRKKRDRKHSKANHGKRPNA
Bbu uL30 MIKRKLRLQLKKARFNASRSRSKN
 * ** * *

The *Bbu* uL30 gene is in operon 171 with 24 other ribosomal proteins as well as other proteins. *Bbu* does not have a known bL37 protein sequence. The *Msm* uL30 gene is in operon 745 with 9 other ribosomal proteins and spans the nucleotides 1,565,535-1,565,720. The *Msm* bL37 gene is not in an operon with other genes and spans the nucleotides 1,998,031-1,998,105 of the *Msm* genome (Genbank ID: CP009494.1). It is not near the *Msm* uL30 protein and its neighboring protein genes in the genome are an anti-sigma factor (CDS: AIU07137.1) and an acetyl-CoA carboxylase (CDS: AIU07138.1). This information is included in the supplementary material section 1.3.

The authors should explain the rationale for doing a 3D classification and whether any masking was used. This could be explained more in the text, results or methods.

The 3D classification was done to find out: (a) if the subunit ratcheting motion was occurring in the hibernating ribosome; (b) if there was a proportion of non-hibernating ribosomes; (c) if the hibernation promoting factor (HPF) protein and the E-tRNA were colocalizing on the 70S ribosome or were a combined density consisting of two different populations; and (d) if there were additional protein densities in sub-populations of the particle images obtained. We did not use masking in the classifications because we did not really find any indications of low occupation densities that were worth pursuing further. We did locally refine the large and small subunit using individual masks, which resulted in better local densities for each. This cleaner map has been deposited along with the original map in the PDB/EMDB deposition (PDB ID: 8FMW, EMD-29298). We have added some more description about this to the manuscript (page 9, paragraph 2, lines 1-6).

The observation regarding H68 is interesting and suggests that the classified large subunit obtained for a sub-population of particles could possibly be an assembly intermediate. A conformational change in h68 was previously observed (i.e. PMC5041480, PMC5145266). The authors could compare these structures as a supplementary to draw conclusions/connections to ribosome assembly. Did the authors check for protein content for these classes? (i.e. any lacking EM-densities?)

We are grateful to the reviewer for pointing out the relevant references about cryo-EM investigations of 50S assembly intermediates that have now been cited in the main text. Also, we have added a figure (Figure S12) in the supplementary material comparing our 50S density to the assembly intermediate densities listed in Table S5. We do not see any missing protein densities in the *Bbu* 50S reconstruction, except for disorder in uL1 and bL7/bL12, which is usually attributable to the flexibility of these regions. This suggests that the *Bbu* 50S subunit structure could be a late 50S assembly intermediate, in which only 23S RNA helices H68 and H69 are yet to be folded correctly. We have added a sentence to the manuscript (page 9, paragraph 3, lines 11-14).

It is probably still possible to identify the putative density for bL38 from the cryo-EM structure. Did the authors check for matching secondary structures over PDBeFold, sequence is less relevant here and the protein hits would be based on the fold of the protein. Another possible solution is to

perform mass spectrometry on the isolated ribosomes and to try and identify the protein from protein hits.

Our efforts to identify bL38 were continuing when we submitted the manuscript to Nature Communications. We have now been able to unambiguously identify the bL38 protein sequence in *Bbu* supported by two lines of evidence: (a) Good fit of sidechains in the density map, (b) 100% protein sequence identification probability by protein mass spectrometry (long gradient LC-MS/MS). It is an annotated protein sequence listed as an uncharacterized 6 kDa protein (55 residues) in *Bbu* with very low homology to the *Flavobacterium johnsoniae* (*Fjo*) bL38 sequence, and directly interacting with the 23S RNA helix 95 containing the sarcin-ricin loop. We have added the details about this identification in the text of the manuscript (page 8, paragraphs 2 and 3 and page 9, paragraph 1), a figure in the manuscript (Figure 5), the methods section in the main manuscript (page 12, paragraph 2), and the supplementary material (Table S7, Figure S7).

For the EM validation, it is best to report bond angles and length to the third decimal and indicate the units in the table. Plots for model vs map and FSC plots and a scheme of the processing should also be indicated together with more information on the data processing process (i.e., total number of micrographs, starting number of particles, etc.).

We have modified Table 2 in the main manuscript and Figure S8 in the supplementary material to include this information.

The computational work on modeling antibiotics is informative but it would have been much more exciting to have visualized the structure with bound antibiotics and explore differences across species.

We agree with this point raised by the reviewer. The computational docking is a proof-of-concept that it is possible to use our *Bbu* ribosome structure for structure-based antibiotic design. We do plan to pursue/report *Bbu* ribosome structures with bound antibiotics in the future.

Reviewer 3.

1) Lines 32-34. The sentence beginning with "The protein bL37" is confusing, especially since *B. burgdorferi* does not produce a bL37 protein. A clearer sentence: "Protein uL30 contains a unique N-terminal alpha-helical extension, which is similar to the bL37 protein that is known only in mycobacterial ribosomes, suggesting that bacterial ribosome proteins bL37 and uL30 may have evolved from a single, longer protein".

We have changed our sentence to the following: 'The protein uL30 contains a unique N-terminal α -helical extension, part of which resembles the bL37 protein seen only in mycobacterial ribosomes, suggesting that bL37 and a shorter uL30 may have evolved from a single, longer form of the uL30 protein.' (Page 2, Abstract, line 12).

2) Lines 47-70. This single paragraph contains three distinct trains of thought. I suggest that it be split into three separate paragraphs, on current line 57 (between "amount each year. and "The ribosome is"), and current line 64 (between "smaller ribosome proteins." and "When diagnosed early").

We have created the three separate paragraphs as suggested (page 3, paragraphs 1-3).

3) Lines 153-154. At this location, please provide an ORF number for the newly-identified bS22 protein. The ORF nomenclature of *B. burgdorferi* type strain B31 will be appropriate.

According to the Spirochete Genome Browser at <http://sgb.leibniz-fli.de>, the bS22 protein is labeled as bb0822. It spans the nucleotide range 867605-867697 in the *Borrelia burgdorferi* B31 chromosome (Genbank ID: AE000783.1). We have added this information to the manuscript along with some additional references about its previous characterization in genome-wide studies (page 6, paragraph 2, line 12-17).

4) Line 158. Please define "Fjo" at this point.

We have defined *Fjo* as *Flavobacterium johnsoniae* at this point as suggested.

5) Lines 162-163. Please state which group/species have the K10O residue change. Although that is in a supplemental table, it would be helpful if this useful piece of information is easily provided to readers, rather than expecting us to go to (and search) another file.

We have added the names of the species with the single residue K10Q change as suggested (page 6, paragraph 2, line 24-25).

6) Line 173. The text says "except for one" - please give us the name of that exception.

We have added the name of the exception as suggested – it is the thermophilic fungus *Chaetomium thermophilum* (*Cth*). The change is in the last line of page 6.

REVIEWERS' COMMENTS

Reviewer #1 (Remarks to the Author):

The authors have made significant changes to the content of the manuscript, taking into account the reviewers' feedback. This includes reorganizing sections, rewriting parts of the text specifically the method sections on modelling.

Based on the thorough improvements made, I wholeheartedly recommend the publication of this manuscript.

Reviewer #2 (Remarks to the Author):

The authors adequately addressed all of my comments. In particular, they were able to identify the putative density for the bL38 protein and expanded their discussion on the observation of a late assembly intermediate from their 3D classifications.

Minor corrections:

In the first part of Line 285, a verb is missing; consider revising: 'It has residues at its N-terminal end not well-resolved, which might interact with the uL14 and bL19 proteins since Bbu bL38 is displaced by 14 Å towards them as compared to the Fjo bL38 protein.'

Table 2: There is a typo - remove the "%" symbol from the values reported for bond angles and lengths.

Reviewer #3 (Remarks to the Author):

The authors have adequately addressed all previous concerns.

Reviewer responses:

Reviewer #1: The authors have made significant changes to the content of the manuscript, taking into account the reviewers' feedback. This includes reorganizing sections, rewriting parts of the text specifically the method sections on modelling. Based on the thorough improvements made, I wholeheartedly recommend the publication of this manuscript.

We thank the reviewer for their evaluation of our revised manuscript, and we are grateful for their positive appraisal.

Reviewer #2: The authors adequately addressed all of my comments. In particular, they were able to identify the putative density for the bL38 protein and expanded their discussion on the observation of a late assembly intermediate from their 3D classifications. Minor corrections: In the first part of Line 285, a verb is missing; consider revising: 'It has 9 residues at its N-terminal end not well-resolved, which might interact with the uL14 and bL19 proteins since Bbu bL38 is displaced by 14 Å... towards them as compared to the Fjo bL38 protein.' Table 2: There is a typo - remove the "%" symbol from the values reported for bond angles and lengths.

We are grateful to the reviewer for their careful consideration and constructive response for our revised manuscript. We have revised the previous statement:

'It has 9 residues at its N-terminal end not well-resolved, which might interact with the uL14 and bL19 proteins since Bbu bL38 is displaced by 14 Å towards them as compared to the Fjo bL38 protein (Figure 5D).'

to

'It has 9 residues at its N-terminal end that are not well-resolved in the cryo-EM density. Since Bbu bL38 is displaced by 14 Å towards the uL14 and bL19 proteins as compared to the Fjo bL38 protein (Fig. 5D), its unresolved N-terminal residues might be interacting dynamically with these proteins.'

We have also removed the '%' symbol for the values reported for bond angles and lengths in Table 2.

Reviewer #3: The authors have adequately addressed all previous concerns.

We appreciate the reviewer's time and effort in assessing our revised manuscript and for their favorable evaluation.